# Using Options and Covariance Testing for Long Horizon Off-Policy Policy Evaluation

**Zhaohan Daniel Guo**
Carnegie Mellon University
Pittsburgh, PA 15213
zguo@cs.cmu.edu

**Philip S. Thomas**
University of Massachusetts Amherst
Amherst, MA 01003
pthomas@cs.umass.edu

**Emma Brunskill**
Stanford University
Stanford, CA 94305
ebrun@cs.stanford.edu

## Abstract

Evaluating a policy by deploying it in the real world can be risky and costly. *Off-policy policy evaluation* (OPE) algorithms use historical data collected from running a previous policy to evaluate a new policy, which provides a means for evaluating a policy without requiring it to ever be deployed. *Importance sampling* is a popular OPE method because it is robust to partial observability and works with continuous states and actions. However, the amount of historical data required by importance sampling can scale exponentially with the *horizon* of the problem: the number of sequential decisions that are made. We propose using policies over temporally extended actions, called *options*, and show that combining these policies with importance sampling can significantly improve performance for long-horizon problems. In addition, we can take advantage of special cases that arise due to options-based policies to further improve the performance of importance sampling. We further generalize these special cases to a general covariance testing rule that can be used to decide which weights to drop in an IS estimate, and derive a new IS algorithm called *Incremental Importance Sampling* that can provide significantly more accurate estimates for a broad class of domains.

## 1 Introduction

One important problem for many high-stakes sequential decision making under uncertainty domains, including robotics, health care, education, and dialogue systems, is estimating the performance of a new policy without requiring it to be deployed. To address this, off-policy policy evaluation (OPE) algorithms use historical data collected from executing one policy (called the behavior policy), to predict the performance of a new policy (called the evaluation policy). Importance sampling (IS) is one powerful approach that can be used to evaluate the potential performance of a new policy [12]. In contrast to model based approaches to OPE [5], importance sampling provides an unbiased estimate of the performance of the evaluation policy. In particular, importance sampling is robust to partial observability, which is often prevalent in real-world domains. Unfortunately, importance sampling estimates of the performance of the evaluation policy can be inaccurate when the horizon of the problem is long: the variance of IS estimators can grow exponentially with the number of sequential decisions made in an episode. This is a serious limitation for applications that involve decisions made over tens or hundreds of steps, like dialogue systems where a conversation might

require dozens of responses, or intelligent tutoring systems that make dozens of decisions about how to sequence the content shown to a student.

Due to the importance of OPE, there have been many recent efforts to improve the accuracy of importance sampling. For example, Dudík et al. [4] and Jiang and Li [7] proposed doubly robust importance sampling estimators that can greatly reduce the variance of predictions when an approximate model of the environment is available. Thomas and Brunskill [16] proposed an estimator that further integrates importance sampling and model-based approaches, and which can greatly reduce mean squared error. These approaches trade-off between the bias and variance of model-based and importance sampling approaches, and result in strongly consistent estimators. Unfortunately, in long horizon settings, these approaches will either create estimates that suffer from high variance or exclusively rely on the provided approximate model, which can have high bias. Other recent efforts that estimate a *value function* using off-policy data rather than just the performance of a policy [6, 11, 19] also suffer from bias if the input state description is not Markovian (such as if the domain description induces partial observability).

To provide better off policy estimates in long horizon domains, we propose leveraging temporal abstraction. In particular, we analyze using options-based policies (policies with temporally extended actions) [14] instead of policies over primitive actions. We prove that the we can obtain an exponential reduction in the variance of the resulting estimates, and in some cases, cause the variance to be independent of the horizon. We also demonstrate this benefit with simple simulations. Crucially, our results can be equivalently viewed as showing that using options can drastically reduce the amount of historical data required to obtain an accurate estimate of a new evaluation policy's performance.

We also show that using options-based policies can result in special cases which can lead to significant reduction in estimation error through dropping importance sampling weights. Furthermore, we generalize the idea of dropping weights and derive a covariance test that can be used to automatically determine which weights to drop. We demonstrate the potential of this approach by constructing a new importance sampling algorithm called Incremental Importance Sampling (INCRIS) and show empirically that it can significantly reduce estimation error.

## 2    Background

We consider an agent interacting with a Markov decision process (MDP) for a finite sequence of time steps. At each time step the agent executes an action, after which the MDP transitions to a new state and returns a real valued reward. Let $s \in S$ be a discrete state, $a \in A$ be a discrete action, and $r$ be the reward bounded in $[0, R_{\max})$.

The transition and reward dynamics are unknown and are denoted by the transition probability $T(s'|s, a)$ and reward density $R(r|s, a)$. A primitive policy maps histories to action probabilities, i.e., $\pi(a_t|s_1, a_1, r_1, \ldots, s_t)$ is the probability of executing action $a_t$ at time step $t$ after encountering history $s_1, a_1, r_1, \ldots, s_t$. The return of a trajectory $\tau$ of $H$ steps is simply the sum of the rewards $G(\tau) = \sum_{t=1}^{H} r_t$. Note we consider the undiscounted setting where $\gamma = 1$. The value of policy $\pi$ is the expected return when running that policy: $V_\pi = \mathbb{E}_\pi(G(\tau))$.

Temporal abstraction can reduce the computational complexity of planning and online learning [2, 9, 10, 14]. One popular form of temporal abstraction is to use sub-policies, in particular options [14]. Let $\Omega$ be the space of trajectories. $o$, an option, consists of $\pi$, a primitive policy (a policy over primitive actions), $\beta : \Omega \to [0, 1]$, a termination condition where $\beta(\tau)$ is the probability of stopping the option given the current partial trajectory $\tau \in \Omega$ from when this option began, and $I \subset S$, an input set where $s \in I$ denotes the states where $o$ is allowed to start. Primitive actions can be considered as a special case of options, where the options always terminate after a single step. $\mu(o_t|s_1, a_1, \ldots, s_t)$ denotes the probability of picking option $o_t$ given history $(s_1, a_1, \ldots, s_t)$ when the previous option has terminated, according to options-based policy $\mu$. A high-level trajectory of length $k$ is denoted by $T = (s_1, o_1, v_1, s_2, o_2, v_2, \ldots, s_k, o_k, v_k)$ where $v_t$ is the sum of the rewards accumulated when executing option $o_t$.

In this paper we will consider batch, offline, off-policy evaluation of policies for sequential decision making domains using both primitive action policies and options-based policies. We will now introduce the general OPE problem using primitive policies: in a later section we will combine this with options-based policies.

In OPE we assume access to historical data, $D$, generated by an MDP, and a behavior policy $\pi_b$. $D$ consists of $n$ trajectories, $\{\tau^{(i)}\}_{i=1}^n$. A trajectory has length $H$, and is denoted by $\tau^{(i)} = (s_1^{(i)}, a_1^{(i)}, r_1^{(i)}, s_2^{(i)}, a_2^{(i)}, r_2^{(i)}, \ldots, s_H^{(i)}, a_H^{(i)}, r_H^{(i)})$. In off-policy evaluation, the goal is to use the data $D$ to estimate the value of an evaluation policy $\pi_e$: $V_{\pi_e}$. As $D$ was generated from running the behavior policy $\pi_b$, we cannot simply use the Monte Carlo estimate. An alternative is to use importance sampling to reweight the data in $D$ to give greater weight to samples that are likely under $\pi_e$ and lesser weight to unlikely ones. We consider per-decision importance sampling (PDIS) [12], which gives the following estimate of the value of $\pi_e$:

$$\text{PDIS}(D) = \frac{1}{n} \sum_{i=1}^n \left( \sum_{t=1}^H \rho_t^{(i)} r_t^{(i)} \right), \qquad \rho_t^{(i)} = \prod_{u=1}^t \frac{\pi_e(a_u^{(i)}|s_u^{(i)})}{\pi_b(a_u^{(i)}|s_u^{(i)})}, \qquad (1)$$

where $\rho_t^{(i)}$ is the weight given to the rewards to correct due to the difference in distribution. This estimator is an unbiased estimator of the value of $\pi_e$:

$$\mathbb{E}_{\pi_e}(G(\tau)) = \mathbb{E}_{\pi_b}(PDIS(\tau)), \qquad (2)$$

where $\mathbb{E}_\pi(\ldots)$ is the expected value given that the trajectories $\tau$ are generated by $\pi$.

For simplicity, hereafter we assume that primitive and options-based policies are a function only of the current state, but our results apply also when the they are a function of the history. Note that importance sampling does not assume that the states in the trajectory are Markovian, and is thus robust to error in the state representation, and in general, robust to partial observability as well.

## 3   Importance Sampling and Long Horizons

We now show how the amount of data required for importance sampling to obtain a good off-policy estimate can scale exponentially with the problem horizon. Notice that in the standard importance sampling estimator, the weight is the product of the ratio of action probabilities. We now prove that this can cause the variance of the policy estimate to be exponential in $H$.[1]

**Theorem 1.** *The mean squared error of the PDIS estimator can be $\Omega(2^H)$.* **Proof.** *See appendix.*

Equivalently, this means that achieving a desired mean squared error of $\epsilon$ can require a number of trajectories that scales exponentially with the horizon. A natural question is whether this issue also arises in a weighted importance sampling [13], a popular (biased) approach to OPE that has lower variance. We show below that the long horizon problem still persists.

**Theorem 2.** *It can take $\Omega(2^H)$ trajectories to shrink the MSE of weighted importance sampling (WIS) by a constant.* **Proof.** *See appendix.*

## 4   Combining Options and Importance Sampling

We will show that one can leverage the advantage of options to mitigate the long horizon problem. If the behavior and evaluation policies are both options-based policies, then the PDIS estimator can be exponentially more data efficient compared to using primitive behavior and evaluation policies.

Due to the structure in options-based policies, we can decompose the difference between the behavior policy and the evaluation policy in a natural way. Let $\mu_b$ be the options-based behavior policy and $\mu_e$ be the options-based evaluation policy. First, we examine the probabilities over the options. The probabilities $\mu_b(o_t|s_t)$ and $\mu_e(o_t|s_t)$ can differ and contribute a ratio of probabilities as an importance sampling weight. Second, the underlying policy, $\pi$, for an option, $o_t$, present in both $\mu_b$ and $\mu_e$ may differ, and this also contributes to the importance sampling weights. Finally, additional or missing options can be expressed by setting the probabilities over missing options to be zero for either $\mu_b$ or $\mu_e$. Using this decomposition, we can easily apply PDIS to options-based policies.

**Theorem 3.** *Let $\mathcal{O}$ be the set of options that have the same underlying policies between $\mu_b$ and $\mu_e$. Let $\overline{\mathcal{O}}$ be the set of options that have changed underlying policies. Let $k^{(i)}$ be the length of the $i$-th high level trajectory from data set $D$. Let $j_t^{(i)}$ be the length of the sub-trajectory produced by option $o_t^{(i)}$. The PDIS estimator applied to $D$ is*

$$PDIS(D) = \frac{1}{n}\sum_{i=1}^{n}\left(\sum_{t=1}^{k^{(i)}} w_t^{(i)} y_t^{(i)}\right) \qquad w_t^{(i)} = \prod_{u=1}^{t} \frac{\mu_e(o_u^{(i)}|s_u^{(i)})}{\mu_b(o_u^{(i)}|s_u^{(i)})}, \qquad (3)$$

$$y_t^{(i)} = \begin{cases} v_t^{(i)} \text{ if } o_t^{(i)} \in \mathcal{O} \\ \sum_{b=1}^{j_t^{(i)}} \rho_{t,b}^{(i)} r_{t,b}^{(i)} \text{ if } o_t^{(i)} \in \overline{\mathcal{O}} \end{cases} \qquad \rho_{t,b}^{(i)} = \prod_{c=1}^{j_t^{(i)}} \frac{\pi_e(a_{t,c}^{(i)}|s_{t,c}^{(i)}, o_t^{(i)})}{\pi_b(a_{t,c}^{(i)}|s_{t,c}^{(i)}, o_t^{(i)})}, \qquad (4)$$

*where $r_{t,b}^{(i)}$ is the b-th reward in the sub-trajectory of option $o_t^{(i)}$ and similarly for s and a.*

*Proof.* This is a straightforward application of PDIS to the options-based policies using the decomposition mentioned. □

Theorem 3 expresses the weights in two parts: one part comes from the probabilities over options which is expressed as $w_t^{(i)}$, and another part comes from the underlying primitive policies of options that have changed with $\rho_{t,b}^{(i)}$. We can immediately make some interesting observations below.

**Corollary 1.** *If no underlying policies for options are changed between $\mu_b$ and $\mu_e$, and all options have length at least J steps, then the worst case variance of PDIS is exponentially reduced from $\Omega(2^H)$ to $\Omega(2^{(H/J)})$*

Corollary 1 follows from Theorem 3. Since no underlying policies are changed, then the only importance sampling weights left are $w_t^{(i)}$. Thus we can focus our attention only on the high-level trajectory which has length at most $H/J$. Effectively, the horizon has shrunk from $H$ to $H/J$, which results in an exponential reduction of the worst case variance of PDIS.

**Corollary 2.** *If the probabilities over options are the same between $\mu_b$ and $\mu_e$, and a subset of options $\overline{\mathcal{O}}$ have changed their underlying policies, then the worst case variance of PDIS is reduced from $\Omega(2^H)$ to $\Omega(2^K)$ where K is an upper bound on the sum of the lengths of the options.*

Corollary 2 follows from Theorem 3. The options whose underlying policies are the same between behavior and evaluation can effectively be ignored, and cut out of the trajectories in the data. This leaves only options whose underlying policies have changed, shrinking down the horizon from $H$ to the length of the leftover options. For example, if only a single option of length 3 is changed, and the option appears once in a trajectory, then the horizon can be effectively reduced to just 3. This result can be very powerful, as the reduced variance becomes independent of the horizon $H$.

## 5 Experiment 1: Options-based Policies

This experiment illustrates how using options-based policies can significantly improve the accuracy of importance-sampling-based estimators for long horizon domains. Since importance sampling is particularly useful when a good model of the domain is unknown and/or the domain involves partial observability, we introduce a partially observable variant of the popular Taxi domain [3] called NoisyTaxi for our simulations (see figure 1).

### 5.1 Partially Observable Taxi

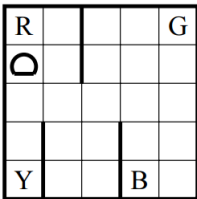

Figure 1: Taxi Domain [3]. It is a $5 \times 5$ gridworld (Figure 1). There are 4 special locations: R,G,B,Y. A passenger starts randomly at one of the 4 locations, and its destination is randomly chosen from one of the 4 locations. The taxi starts randomly on any square. The taxi can move one step in any of the 4 cardinal directions N,S,E,W, as well as attempt to pickup or drop off the passenger. Each step has a reward of $-1$. An invalid pickup or dropoff has a $-10$ reward and a successful dropoff has a reward of 20.

In NoisyTaxi, the location of the taxi and the location of the passenger is partially observable. If the row location of the taxi is $c$, the agent observes $c$ with probability 0.85, $c + 1$ with probability 0.075

and $c - 1$ with probability 0.075 (if adding or subtracting 1 would cause the location to be outside the grid, the resulting location is constrained to still lie in the grid). The column location of the taxis is observed with the same noisy distribution. Before the taxi successfully picks up the passenger, the observation of the location of the passenger has a probability of 0.15 of switching randomly to one of the four designated locations. After the passenger is picked up, the passenger is observed to be in the taxi with 100% probability (e.g. no noise while in the taxi).

## 5.2 Experimental Results

We consider $\epsilon$-greedy option policies, where with probability $1 - \epsilon$ the policy samples the optimal option, and probability $\epsilon$ the policy samples a random option. Options in this case are $n$-step policies, where "optimal" options involve taking $n$-steps of the optimal (primitive action) policy, and "random" options involve taking $n$ random primitive actions.[2] Our behavior policies $\pi_b$ will use $\epsilon = 0.3$ and our evaluation policies $\pi_e$ use $\epsilon = 0.05$. We investigate how the accuracy of estimating $\pi_e$ varies as a function both of the number of trajectories and the length of the options $n = 1, 2, 3$. Note $n = 1$ corresponds to having a primitive action policy.

Empirically, all behavior policies have essentially the same performance. Similarly all evaluation policies have essentially the same performance. We first collect data using the behavior policies, and then use PDIS to evaluate their respective evaluation policies.

Figure 2 compares the MSE (log scale) of the PDIS estimators for the evaluation policies.

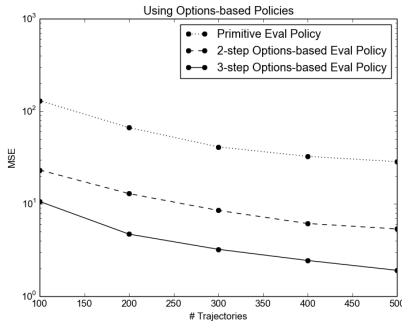

Figure 2: Comparing the MSE of PDIS between primitive and options-based behavior and evaluation policy pairs. Note the y-axis is a log scale. Our results show that PDIS for the options-based evaluation policies are an order of magnitude better than PDIS for the primitive evaluation policy. Indeed, Corollary 1 shows that the $n$-step options policies are effectively reducing the horizon by a factor of $n$ over the primitive policy. As expected, the options-based policies that use 3-step options have the lowest MSE.

## 6 Going Further with Options

Often options are used to achieve a specific sub-task in a domain. For example in a robot navigation task, there may be an option to navigate to a special fixed location. However one may realize that there is a faster way to navigate to that location, so one may change that option and try to evaluate the new policy to see whether it is actually better. In this case the old and new option are both always able to reach the special location; the only difference is that the new option could get there faster. In such a case we can further reduce the variance of PDIS. We now formally define this property.

**Definition 1.** *Given behavior policy $\mu_b$ and evaluation policy $\mu_e$, an option $o$ is called **stationary**, if the distribution of the states on which $o$ terminates is always the same for $\mu_b$ and $\mu_e$. The underlying policy for option $o$ can differ for $\mu_b$ and $\mu_e$; only the termination state distribution is important.*

A stationary option may not always arise due to solving a sub-task. It can also be the case that a stationary option is used as a way to perform a soft reset. For example, a robotic manipulation task may want to reset arm and hand joints to a default configuration in order to minimize sensor/motor error, before trying to grasp a new object.

Stationary options allows us to point to a step in a trajectory where we know the state distribution is fixed. Because the state distribution is fixed, we can partition the trajectory into two parts. The beginning of the second partition would then have state distribution that is independent of the actions

chosen in the first partition. We can then independently apply PDIS to each partition, and sum up the estimates. This is powerful because it can halve the effective horizon of the problem.

**Theorem 4.** *Let $\mu_b$ be an options-based behavior policy. Let $\mu_e$ be an options-based evaluation policy. Let $\mathcal{O}$ be the set of options that $\mu_b$ and $\mu_e$ use. The underlying policies of the options in $\mu_e$ may be arbitrarily different from $\mu_b$.*

*Let $o_1$ be a stationary option. We can decompose the expected value as follows. Let $\tau_1$ be the first part of a trajectory up until and including the first occurrence of $o_1$. Let $\tau_2$ be the part of the trajectory after the first occurrence of $o_1$ up to and including the first occurrence of $o_2$. Then*

$$\mathbb{E}_{\mu_e}(G(\tau)) = \mathbb{E}_{\mu_b}(PDIS(\tau)) = \mathbb{E}_{\mu_b}(PDIS(\tau_1)) + \mathbb{E}_{\mu_b}(PDIS(\tau_2)) \tag{5}$$

**Proof.** *See appendix.*

Note that there are no conditions on how the probabilities over options may differ, nor on how the underlying policies of the non-stationary options may differ. This means that, regardless of these differences, the trajectories can be partitioned and PDIS can be independently applied. Furthermore, Theorem 3 can still be applied to each of the independent applications of PDIS. Combining Theorem 4 and Theorem 3 can lead to more ways of designing a desired evaluation policy that will result in a low variance PDIS estimate.

## 7    Experiment 2: Stationary Options

We now demonstrate Theorem 4 empirically on NoisyTaxi. In NoisyTaxi, we know that a primitive $\epsilon$-greedy policy will eventually pick up the passenger (though it may take a very long time depending on $\epsilon$). Since the starting location of the passenger is uniformly random, the location of the taxi immediately after picking up the passenger is also uniformly random, but over the four pickup locations. This implies that, regardless of the $\epsilon$ value in an $\epsilon$-greedy policy, we can view executing that $\epsilon$-greedy policy until the passenger is picked up as a new "PickUp-$\epsilon$" option that always terminates in the same state distribution.

Given this argument, we can use Theorem 4 to decompose any NoisyTaxi trajectory into the part before the passenger is picked up, and the part after the passenger is picked up, estimate the expected reward for each, and then sum. As picking up the passenger is often the halfway point in a trajectory (depending on the locations of the passenger and the destination), we can perform importance sampling over two, approximately half length, trajectories. More concretely, we consider two $n = 1$ options (e.g. primitive action) $\epsilon$-greedy policies. Like in the prior subsection, the behavior policy has $\epsilon = 0.3$ and the evaluation policy has $\epsilon = 0.05$. We compare performing normal PDIS to estimate the value of the evaluation policy to estimating it using partitioned-PDIS using Theorem 4. See Figure 3 for results.

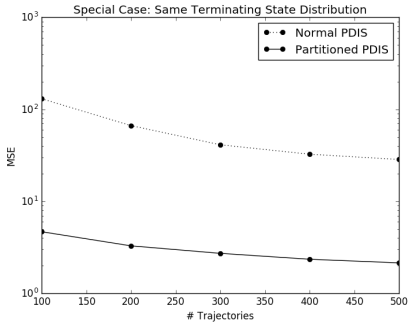

Figure 3: Comparing MSE of Normal PDIS and PDIS that uses Theorem 4. We gain an order of an order of magnitude reduction in MSE (labeled Partitioned-PDIS). Note this did not require that the primitive policy used options: we merely used the fact that if there are subgoals in the domain where the agent is likely to go through with a fixed state distribution, we can leverage that to decompose the value of a long horizon into the sum over multiple shorter ones. Options is one common way this will occur, but as we see in this example, this can also occur in other ways.

## 8    Covariance Testing

The special case of stationary options can be viewed as a form of dropping certain importance sampling weights from the importance sampling estimator. With stationary options, the weights before the stationary options are dropped when estimating the rewards thereafter. By considering

the bias incurred when dropping weights, we derive a general rule involving covariances as follows. Let $W_1 W_2 r$ be the ordinary importance sampling estimator for reward $r$ where the product of the importance sampling weights are partitioned into two products $W_1$ and $W_2$ using some general partitioning scheme such that $\mathbb{E}(W_1) = 1$. Note that this condition is satisfied when $W_1, W_2$ are chosen according to commonly used schemes such as fixed timesteps (not necessarily consecutive) or fixed states, but can be satisfied by more general schemes as well. Then we can consider dropping the product of weights $W_1$ and simply output the estimate $W_2 r$:

$$\mathbb{E}(W_1 W_2 r) = \mathbb{E}(W_1)\mathbb{E}(W_2 r) + \text{Cov}(W_1, W_2 r) \tag{6}$$
$$= \mathbb{E}(W_2 r) + \text{Cov}(W_1, W_2 r) \tag{7}$$

This means that if $\text{Cov}(W_1, W_2 r) = 0$, then we can drop the weights $W_1$ with no bias. Otherwise, the bias incurred is $\text{Cov}(W_1, W_2 r)$. Then we are free to choose $W_1, W_2$ to balance the reduction in variance and the increase in bias.

## 8.1 Incremental Importance Sampling (INCRIS)

Using the Covariance Test (eqn 7) idea, we propose a new importance sampling algorithm called Incremental Importance Sampling (INCRIS). This is a variant of PDIS where for a reward $r_t$, we try to drop all but the most recent $k$ importance sampling weights, using the covariance test to optimize $k$ in order to lower MSE.

Let $\pi_b$ and $\pi_e$ be the behavior and evaluation policies respectively (they may or may not be options-based policies). Let $D = \{\tau^{(1)}, \tau^{(2)}, \dots, \tau^{(n)}\}$ be our historical data set generated from $\pi_b$ with $n$ trajectories of length $H$. Let $\rho_t = \frac{\pi_e(a_t|s_t)}{\pi_b(a_t|s_t)}$. Let $\rho_t^{(i)}$ be the same but computed from the $i$-th trajectory. Suppose we are given estimators for covariance and variance. See algorithm 1 for details.

---

**Algorithm 1** INCRIS

1: **Input:** $D$
2: **for** $t = 1$ to $H$ **do**
3:     **for** $k = 0$ to $t$ **do**
4:         $A_k = \prod_{j=1}^{t-k} \rho_j$
5:         $B_k = \prod_{j=t-k+1}^{t} \rho_j$
6:         Estimate $\text{Cov}(A_k, B_k r_t)$ and denote $\widehat{C}_k$
7:         Estimate $\text{Var}(B_k r_t)$ and denote $\widehat{V}_k$
8:         Estimate MSE with $\widehat{MSE}_k = \widehat{C}_k^2 + \widehat{V}_k$
9:     **end for**
10:    $k' = \text{argmin}_k \widehat{MSE}_k$
11:    Let $\widehat{r_t} = \frac{1}{n}\sum_{i=1}^{n} B_{k'}^{(i)} r_t$
12: **end for**
13: **Return** $\sum_{t=1}^{H} \widehat{r_t}$

---

## 8.2 Strong Consistency

In the appendix, we provide a proof that INCRIS is strongly consistent. We now give a brief intuition for the proof. As $n$ goes to infinity, the estimates for the MSE get better and better and converge to the bias. We know that if we do not drop any weights, we get an unbiased estimate and thus the smallest MSE estimate will go to zero. Thus we get more and more likely to pick $k$ that correspond to unbiased estimates.

## 9 Experiment 3: Incremental Importance Sampling

To evaluate INCRIS, we constructed a simple MDP that exemplifies to properties of domains for which we expect INCRIS to be useful. Specifically, we were motivated by the applications of reinforcement learning methods to type 1 diabetes treatments [1, 17] and digital marketing applications [15]. In these applications there is a natural place where one might divide data into episodes: for type 1

diabetes treatment, one might treat each day as an independent episode, and for digital marketing, one might treat each user interaction as an independent episode.

However, each day is not actually independent in diabetes treatment—a person's blood sugar in the morning depends on their blood sugar at the end of the previous day. Similarly, in digital marketing applications, whether or not a person clicks on an ad might depend on which ads they were shown previously (e.g., someone might be less likely to click an ad that they were shown before and did not click on then). So, although this division into episodes is reasonable, it does not result in episodes that are completely independent, and so importance sampling will not produce consistent estimates (or estimates that can be trusted for high-confidence off-policy policy evaluation [18]). To remedy this, we might treat all of the data from a single individual (many days, and many page visits) as a single episode, which contains nearly-independent subsequences of decisions.

To model this property, we constructed an MDP with three states, $s_1$, $s_2$, and $s_3$ and two actions, $a_1$ and $a_2$. The agent always begins in $s_1$, where taking action $a_1$ causes a transition to $s_2$ with a reward of $+1$ and taking action $a_2$ causes a transition to $s_3$ with a reward of $-1$. In $s_2$, both actions lead to a terminal absorbing state with reward $-2 + \epsilon$, and in $s_3$ both actions lead to a terminal absorbing state with reward $+2$. For now, let $\epsilon = 0$. This simple MDP has a horizon of 2 time steps—after two actions the agent is always in a terminal absorbing state. To model the aforementioned examples, we modified this simple MDP so that whenever the agent would transition to the terminal absorbing state, it instead transitions back to $s_1$. After visiting $s_1$ fifty times, the agent finally transitions to a terminal absorbing state. Furthermore, to model the property that the fifty sub-episodes within the larger episode are not completely independent, we set $\epsilon = 0$ initially, and $\epsilon = \epsilon + 0.01$ whenever the agent enters $s_2$. This creates a slight dependence across the sub-episodes.

For this illustrative domain, we would like an importance sampling estimator that assumes that sub-episodes are independent when there is little data in order to reduce variance. However, once there is enough data for the variances of estimates to be sufficiently small relative to the bias introduced by assuming that sub-episodes are independent, the importance sampling estimator should automatically begin considering longer sequences of actions, as INCRIS does. We compared INCRIS to ordinary importance sampling (IS), per-decision importance sampling (PDIS), weighted importance sampling (WIS), and consistent weighted per-decision importance sampling (CWPDIS). The behavior policy selects actions randomly, while the evaluation policy selects action $a_1$ with a higher probability than $a_2$. In Figure 4 we report the mean squared errors of the different estimators using different amounts of data.

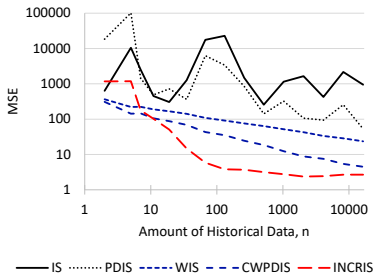

Figure 4: Performance of different estimators on the simple MDP that models properties of the diabetes treatment and digital marketing applications. The reported mean squared errors are the sample mean squared errors from 128 trials. Notice that INCRIS achieves an order of magnitude lower mean squared error than all of the other estimators, and for some $n$ it achieves two orders of magnitude improvement over the unweighted importance sampling estimators.

## 10   Conclusion

We have shown that using options-based behavior and evaluation policies allow for lower mean squared error when using importance sampling due to their structure. Furthermore, special cases may naturally arise when using options, such as when options terminate in a fixed state distribution, and lead to greater reduction of the mean squared error.

We examined options as a first step, but in the future these results may be extended to full hierarchical policies (like the MAX-Q framework). We also generalized naturally occurring special cases with covariance testing that leads to dropping out weights in order to improve importance sampling predictions. We showed an instance of covariance testing in the algorithm INCRIS, which can greatly improve estimation accuracy for a general class of domains, and hope to derive more powerful estimators based on covariance testing that can apply to even more domains in the future.

## Acknowledgements

The research reported here was supported in part by an ONR Young Investigator award, an NSF CAREER award, and by the Institute of Education Sciences, U.S. Department of Education. The opinions expressed are those of the authors and do not represent views of NSF, IES or the U.S. Dept. of Education.

## Footnotes

[1]These theorems can be seen as special case instantiations of Theorem 6 in [8] with simpler, direct proofs.

[2]We have also tried using more standard options that navigate to a specific destination, and the experiment results closely mirror those shown here.

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
