[Supplementary Material]

# Appendix: Using Options and Covariance Testing for Long Horizon Off-Policy Policy Evaluation

## A Proof of Theorem 1

Because PDIS is an unbiased estimator of an evaluation policy's performance, its MSE is equal to its variance. To prove the theorem statement, we provide an existence proof by constructing a sample MDP where, given a particular behavior policy, there is an evaluation policy whose estimate under PDIS will have a variance that scales exponentially with the horizon $H$.

Consider a discrete state and action Markov decision process. The horizon is $H$ and the MDP has $2H + 1$ states and 2 actions. The states form two chains: the top chain has length $H + 1$ and the bottom chain has length $H$. Label the states of the top chain as $x_1, \ldots, x_{H+1}$, and the states of the bottom chain be $y_1, \ldots, y_H$. The start state is $x_1$. An episode halts after $H$ steps. The two actions are $a_1, a_2$. Taking action $a_1$ in the top chain deterministically transitions to the next state in the top chain i.e. from $x_i$ to $x_{i+1}$. Taking action $a_2$ in the top chain deterministically transitions to the corresponding state in the bottom chain i.e. from $x_i$ to $y_i$. The reward is zero everywhere except a reward of 1 is received for executing action $a_1$ at state $x_H$. The optimal policy is to always take action $a_1$.

Let the behavior policy $\pi_b$ be uniformly random i.e. there is always a probability of $1/2$ of picking either action. The evaluation policy is the optimal policy, $\pi_e(s) = a_1$ for all states.

Since the only nonzero reward is the single reward of 1 at $x_H$ for action $a_1$, and it is only possible to reach that state by taking action $a_1$ for $H$ steps, PDIS reduces to a sum only over trajectories consisting solely of $H$ steps of action $a_1$, whose weights are $\rho = \prod_{u=1}^{H} \frac{\pi_e(a_1|s_u^{(i)})}{\pi_b(a_1|s_u^{(i)})} = 2^H$. The PDIS estimate of the evaluation policy is a scaled Binomial distribution where with probability $p = \frac{1}{2^H}$ a trajectory's weighted return is $2^H$ and zero otherwise. Thus the variance of the PDIS estimate of $\pi_e$ is $\frac{1}{n}(2^H - 1)$ for $n$ trajectories, which is $\Omega(2^H)$ .

## B Proof of Theorem 2

We prove the above statement by constructing a MDP and selecting a behavior and target policy which will result in the stated MSE dependence on the horizon. We consider the same MDP, $\pi_b$, and $\pi_e$ as used in the proof above. For this particular MDP, the only weight that matters is the weight associated with the single final reward of the correct trajectory, so per-decision importance sampling and ordinary importance sampling are equivalent.

If the optimal trajectory does not appear in the historical data, then WIS is undefined. This is because the weight of any nonoptimal trajectory is 0, so dividing by the sum of the weights is undefined. However if we change $\pi_e$ from deterministically picking $a_1$ to picking $a_1$ with arbitrarily high probability, then the weights of nonoptimal trajectories will be arbitrarily close to zero, resulting in a WIS estimate of 0. Thus we define WIS to estimate a value of 0 when the optimal trajectory does not appear in the data. Any optimal trajectory that appears in your data will have a weight of $2^H$. Then because the weights of nonoptimal trajectories are 0, the WIS estimate will be exactly 1. Thus as soon as WIS sees at least one correct trajectory it will have

the perfect estimate, otherwise the estimate will be 0. The WIS estimate is a Bernoulli distribution where the probability of 1 is the probability of at least one optimal trajectory appearing in the data.

Since the WIS estimate is Bernoulli, its variance is bounded by a constant. Furthermore the variance is small. Thus we take a closer look at the bias, since MSE is the sum of the variance and bias squared. First we compute the probability the WIS returns 1. This is the probability of at least one optimal trajectory appearing, which is equivalent to one minus the probability of no optimal trajectory appearing: $1 - \left(1 - \frac{1}{2^H}\right)^n$. Thus the expected value of the WIS estimate is $1 - \left(1 - \frac{1}{2^H}\right)^n$. Then the bias is $\left(1 - \frac{1}{2^H}\right)^n$. Let the bias be $B$. We will compute how much data is needed to compensate for the increase in the bias when $H$ increases. Rearranging and solving for $n$ (using a taylor approximation) we get $n = \frac{\log B}{\log\left(1 - \frac{1}{2^H}\right)} \approx \frac{\log B}{-\frac{1}{2^H}} \approx \Omega(2^H)$. Thus we need an exponential number of trajectories to compensate for the increase in bias when the horizon $H$ is increased. Since MSE consists partly of biased squared, we would need even more data to compensate for a squared increase in bias, but for simplicity we still use an exponential bound.

# C   Proof of Theorem 4

Let $t^*$ be the timestep when $o_1$ terminates. Then

$$\mathbb{E}_{\mu_e}(J(\tau)) \tag{1}$$

$$= \mathbb{E}_{\mu_e}(J(\tau_1) + J(\tau_2)) \tag{2}$$

$$= \mathbb{E}_{\mu_e}(J(\tau_1)) + \mathbb{E}_{\mu_e}(J(\tau_2)) \tag{3}$$

$$= \mathbb{E}_{\mu_e}(J(\tau_1)) + \mathbb{E}_{\mu_e}\left(\mathbb{E}_{\mu_e}(J(\tau_2)|s_{t^*} = s)\right) \tag{4}$$

$$= \mathbb{E}_{\mu_e}(J(\tau_1)) + \sum_{s \in \mathcal{S}} \Pr(s_{t^*} = s|\mu_e)\left(\mathbb{E}_{\mu_e}(J(\tau_2)|s_{t^*} = s)\right) \tag{5}$$

$$= \mathbb{E}_{\mu_b}(PDIS(\tau_1)) + \sum_{s \in \mathcal{S}} \Pr(s_{t^*} = s|\mu_e)\left(\mathbb{E}_{\mu_b}(PDIS(\tau_2)|s_{t^*} = s)\right) \tag{6}$$

$$= \mathbb{E}_{\mu_b}(PDIS(\tau_1)) + \sum_{s \in \mathcal{S}} \Pr(s_{t^*} = s|\mu_b)\left(\mathbb{E}_{\mu_b}(PDIS(\tau_2)|s_{t^*} = s)\right) \tag{7}$$

$$= \mathbb{E}_{\mu_b}(PDIS(\tau_1)) + \mathbb{E}_{\mu_b}\left(\mathbb{E}_{\mu_b}(PDIS(\tau_2)|s_{t^*} = s)\right) \tag{8}$$

$$= \mathbb{E}_{\mu_b}(PDIS(\tau_1)) + \mathbb{E}_{\mu_b}(PDIS(\tau_2)) \tag{9}$$

where eqn 4 follows from the law of total expectation, eqn 6 follows from using PDIS with a fixed initial state distribution $s_{t^*} = s$, eqn 7 follows because $s_{t^*}$ is the terminating state for option $o_1$ whose terminating state distribution stayed the same between $\mu_b$ and $\mu_e$, and eqn 9 follows from the law of total expectation.

# D   Strong Consistency of INCRIS

Given strongly consistent estimators for covariance and variance (e.g. sample covariance and sample variance), we show that INCRIS is consistent by showing that $\sum_{t=1}^{H} \hat{r}_t \xrightarrow{\text{a.s.}} \mathbb{E}_{\pi_e}\left(\sum_{t=1}^{H} r_t\right)$, i.e. the total expected value under the evaluation policy. Since we have a finite sum, it is sufficient to show that for all $t$, $\hat{r}_t \xrightarrow{\text{a.s.}} \mathbb{E}_{\pi_e}(r_t)$.

For any $k$, because we have strongly consistent covariance and variance estimators, as $n \to \infty$ we have that $\widehat{V_k} \xrightarrow{\text{a.s.}} \text{Var}(B_k r_t)$ and since $\text{Var}(B_k r_t)$ converges to zero, we also have that $\widehat{V_k} \xrightarrow{\text{a.s.}} 0$. Similarly, $\widehat{C_k} \xrightarrow{\text{a.s.}} \text{Cov}(A_k, B_k r_t)$. Therefore $\widehat{MSE}_k \xrightarrow{\text{a.s.}} (\text{Cov}(A_k, B_k r_t))^2$.

By eqn. **??** we have that $\text{Cov}(A_k, B_k r_t) = \mathbb{E}(A_k B_k r_t) - \mathbb{E}(B_k r_t)$, which is the bias of $\mathbb{E}(A_k r_t)$.

Let $K^* = \{k | \text{Cov}(A_k, B_k r_t) = 0\}$. Notice that $t \in K^*$ since $\hat{r}_t = \frac{1}{n} \sum_{i=1}^{n} \prod_{j=1}^{t} \rho_j^{(i)} r_t$ is the ordinary importance sampling estimator, which is unbiased.

We want to show that as $n \to \infty$, the algorithm eventually picks $k' \in K^*$ and so $\widehat{r}_t$ is an unbiased estimate. To do so, let $(\Omega, \Sigma, p)$ be the probability space. Since for all $k$, $\widehat{MSE}_k \xrightarrow{\text{a.s.}} (\text{Cov}(A_k, B_k r_t))^2$, then $p(G) = 1$ where $G = \{\omega \in \Omega | \forall k \lim_{n\to\infty} \widehat{MSE}_k = (\text{Cov}(A_k, B_k r_t))^2\}$.

Now we can restrict our focus to only events $\omega \in G$. First, let

$$\epsilon_{gap} = \min_{k | \text{Cov}(A_k, B_k r_t) > 0} (\text{Cov}(A_k, B_k r_t))^2$$

which is the smallest nonzero MSE. Since $\lim_{n\to\infty} \widehat{MSE}_k = (\text{Cov}(A_k, B_k r_t))^2$, by definition of limit, we have that there exists $n_0$ such that for all $n \geq n_0$, $|\widehat{MSE}_k - (\text{Cov}(A_k, B_k r_t))^2| < \frac{\epsilon_{gap}}{3}$. By the definition of $\epsilon_{gap}$, for $n \geq n_0$, $k' \in K^*$.

We have shown that the algorithm eventually picks $k' \in K^*$. Next we will show that this implies $\widehat{r}_t \xrightarrow{\text{a.s.}} \mathbb{E}_{\pi_e}(r_t)$.

By the strong law of large numbers, for any $k$, $\left(\frac{1}{n}\sum_{i=1}^{n} B_k^{(i)} r_t\right) \xrightarrow{\text{a.s.}} \mathbb{E}(B_k r_t)$. Then we know that $p(G') = 1$ where $G' = \{\omega \in \Omega | \lim_{n\to\infty} \left(\frac{1}{n}\sum_{i=1}^{n} B_k^{(i)} r_t\right) = \mathbb{E}(B_k r_t)\}$. Then $p(G \cap G') = 1$, and we can restrict our focus to $\omega \in (G \cap G')$. We have already shown that when $\omega \in (G \cap G')$, there exists $n_0$ such that for all $n \geq n_0$, $k' \in K^*$. But we also know that when $\omega \in (G \cap G')$, $\lim_{n\to\infty} \left(\frac{1}{n}\sum_{i=1}^{n} B_{k'}^{(i)} r_t\right) = \mathbb{E}(B_{k'} r_t)$ for any $k'$, so for $k' \in K^*$, $\lim_{n\to\infty} \left(\frac{1}{n}\sum_{i=1}^{n} B_{k'}^{(i)} r_t\right) = \mathbb{E}(B_{k'} r_t) = \mathbb{E}(B_t r_t) = \mathbb{E}_{\pi_e}(r_t)$. Therefore, for all $\epsilon > 0$, we will always be able to find some $n_0$ large enough such that for all $n \geq n_0$, when $\omega \in (G \cap G')$, we have $k' \in K^*$ and $\widehat{r}_t = \left(\frac{1}{n}\sum_{i=1}^{n} B_{k'}^{(i)} r_t\right)$ and therefore $|\widehat{r}_t - \mathbb{E}(B_t r_t)| < \epsilon$.

Thus we have shown $\widehat{r}_t \xrightarrow{\text{a.s.}} \mathbb{E}_{\pi_e}(r_t)$, and so INCRIS is strongly consistent.

# References