[Reviews · NeurIPS 2017]

Reviewer 1



The paper addresses off-policy policy evaluation: determining the value of a policy using experience obtained by following a different policy. The authors build on existing work on importance sampling, focusing on how temporally-extended actions (options) can be useful in this setting, especially when the horizon is long. The specific contributions of the paper are as follows: (1) importance-sampling estimator in the presence of options, discussion of how it can reduce estimation error compared to when using only primitive actions, experimental confirmation in the taxi domain, (2) importance sampling in the presence of an option with a fixed distribution of terminal states, discussion of how this can reduce estimation error compared to when such an option is not present, experimental confirmation in the taxi domain, (3) a general-purpose, incremental importance-sampling algorithm that can be used with or without options, along with an experimental evaluation in a simple (but well-motivated) MDP. This last algorithm drops distant experiences, examining covariant structure to determine a numerical value for "distant". The problem addressed in the paper is important, with many real-world application areas. The results are straightforward but potentially useful. How useful they turn out to be depends on how often real-world problems exhibit the various forms of structure exploited by the authors. The domains used in the paper are well motivated but synthetic. Real-world domains and data sets, especially in experiment 3, would go a long way in making a stronger case for the approach. The paper is clear and well organized. Minor comment: In experiment 1, it would be informative to see additional conditions tested, using for example options that are typical for the taxi domain (for instance, driving to the passenger location). The options tested by the authors are informative but artificial (they simply follow the optimal policy from a given state for a given number of steps).

Reviewer 2



The authors investigate how options influence the variance of importance sampling estimators to increase the length of trajectories that off-policy evaluation approaches can be applied to. They prove that for off-policy evaluation with WIS and single timestep actions it can take an exponential number of trajectories in the horizon to achieve a desired level of accuracy. This is particularly problematic because WIS has much lower variance than IS. Corollaries 1 and 2 prime the reader’s intuition about how options can reduce impact of long horizons by describing two special cases. The authors introduce the notion of stationary options and demonstrate how the trajectory can be decomposed into the importance weighted return up to the first occurrence of the stationary option and the remainder of the trajectory. Unfortunately, the condition for stationary options is difficult to enforce in practice. So the authors introduce a covariance test that helps to relax this assumption (by introducing some bias). INCRIS exploits the covariance test to drop importance sampling weights when doing so minimizes an estimate of the MSE. The authors should spend more time describing INCRIS. Moving from the covariance test to the algorithm is too abrupt and it is hard to relate this back to the topic of off-policy evaluation over option policies. It is confusing in Corollaries 1 and 2 that K is used for two different things (number of decision points and total number of changed steps). Line 66 says ‘ldots’ which should be …

Reviewer 3



[I have read the other reviews and the author feedback, and updated my rating to be borderline accept. The paper will be substantially stronger with more realistic expts. In the absence of such expts, I suspect that the IncrIS estimator will run into substantial computational challenges and sub-optimal MSE in realistic scenarios.] The paper introduces new off-policy estimators to evaluate MDP policies. The options-based estimator of Theorem3 is a straightforward adaptation of per-step importance sampling. Section6 makes an interesting observation: if at any step in the trajectory we know that the induced state distribution is the same between the behavior policy and the evaluation policy, we can partition the trajectory into two sub-parts and conduct per-step importance sampling on each sub-part. This observation may prove to be more generally useful; in this paper, they exploit this observation to propose the IncrIS estimator. IncrIS uses sample variance and covariance estimates (presumably; this detail is missing in their experiment description) to detect what prefix of sampled trajectories can be safely ignored [i.e. yields smaller (estimated) MSE of per-step importance sampling on the resulting suffix]. This is an interesting approach to bias-variance trade-offs in importance-sampling-estimator design that can be explored further; e.g. covariance testing requires us to fix the choice of k (ignored prefix length) across the entire set of trajectories. Can we condition on the observed trajectories to adaptively discard importance weights? In summary, the contributions up to Section 6 seem very incremental. IncrIS introduces an intriguing bias-variance trade-off but the experimental evidence (MSE in a constructed toy MDP)is not very convincing. In realistic OPE settings, it is likely that horizons will be large (so IncrIS can shine), but there may be lots of sampled trajectories (computational challenge for IncrIS to compute \hat{MSE}), and we almost certainly won't be in the asymptotic regime where we can invoke strong consistency of IncrIS (so, \hat{MSE} may yield severely sub-optimal choices, and IncrIS may have poor empirical MSE). Due to these intuitions, a more realistic experiment will be much more informative in exploring the strengths and weaknesses of the proposed estimators. Minor: 66: \ldots 112: one *can* leverage *the* advantage?